# Prevalence and Antimicrobial Susceptibility Profiles of Microorganisms Associated with Lower Reproductive Tract Infections in Women from Southern Poland—Retrospective Laboratory-Based Study

**DOI:** 10.3390/ijerph18010335

**Published:** 2021-01-05

**Authors:** Jadwiga Wójkowska-Mach, Monika Pomorska-Wesołowska, Małgorzata Romanik, Dorota Romaniszyn

**Affiliations:** 1Department of Microbiology, Faculty of Medicine, Jagiellonian University Medical College, 31-008 Krakow, Poland; varicella_zoster@interia.pl or; 2Department of Microbiology, Analytical and Microbiological Laboratory of KORLAB NZOZ, 41-700 Ruda Śląska, Poland; monikapw@op.pl; 3Department of Medical Microbiology, Medical University of Silesia, 40-055 Katowice, Poland; romanikmargo@poczta.onet.pl or

**Keywords:** vaginal infections, lower genital tract, *Escherichia coli*, *Candida* spp.

## Abstract

Objective: Female infections affecting the genital tract include sexually transmitted diseases, endogenous infections such as vulvovaginal candidiasis, bacterial vaginosis (BV) or aerobic vaginitis (AV) and healthcare-associated infections. The aim of the study was to analyze the etiological factors of the vaginal dysbacteriosis, and the antimicrobial susceptibility of the dominant bacterial and fungal infections in different age groups of outpatient women from the Silesian Region. Materials and methods: A retrospective laboratory-based multi-center study encompassed 4994 women of different ages in Silesian Voivodeship, in the south of Poland; patients who had vaginal swabs collected as per physicians’ orders during the period from 1 January 2017 until 30 June 2018 were included in the study. The inclusion criteria were: non-hospitalized female, aged ≤80, with suspected vulvovaginal candidiasis or bacterial vaginosis and clinical sings of infections. Results: Gram-positive cocci were the ones most often isolated: *Enterococcus faecalis* (29.2%) and *Streptoccoccus agalactiae* (13.1%), followed by bacilli from the Enterobacteriaceae group, including *Escherichia coli* (26.3%). The presence of *Streptococcus agalactiae* was confirmed in 13.1%, slightly more often in the 45–80 age group, and *Gardnerella vaginalis* in 6.4%, most often in women aged 15–24. The prevalence of yeast-like infections was 24.3%, *Candida albicans* accounted for 78.3%, whereas among *C. non-albicans* spp.—*C. glabrata* dominated (14.9%) followed by *C. parapsilosis* (3.8%). The highest resistance was observed only in Streptococcus agalactiae as the MLSB mechanism (Macrolide-lincosamide-streptogramin B) was identified in 38.6% of strains. The prevalence of vulvovaginal candidiasis was 24.3%, the highest in women aged 15–44. Conclusions: Drug resistance in studied vulvovaginitis was associated only with *Streptococcus agalactiae*. A high proportion of yeast-like aetiology was found, probably associated with recurrent infections. In the analyzed cases only the Amsel criteria and culture methods were used for diagnosis without preparations and microbiological Nugent criteria.

## 1. Introduction

The bacterial flora in the vagina of a healthy woman of reproductive age is dominated by lactobacilli. Besides of *Lactobacillus* species, the following bacterial genera may also be isolated from the female genital tract: *Streptococcus, Staphylococcus, Enterococcus, Escherichia* and *Klebsiella,* as well as *Candida* yeasts. Female infections affecting the genital tract include sexually transmitted diseases, endogenous infections such as vulvovaginal candidiasis, bacterial vaginosis (BV) or aerobic vaginitis (AV) and healthcare-associated infections [1,2].

In a healthy woman, vaginal microorganisms are mutually antagonistic and interdependent, maintaining a dynamic balance regulated by the endocrine system and the local immune system and influenced by the internal environment of the vagina [3]. The composition of microbiota may vary considerably depending on the age of the woman, hormone management, sexual activity, as well as exposure to hygienic substances used in vaginal irrigation.

In developed countries, overgrowth of endogenous bacteria normally found in the vagina is usually the most common cause of vaginal discharge in women of reproductive age. In this case of bacterial dysbiosis a shift in vaginal microbiota from lactobacilli domination to an increased growth of mixed anaerobes in BV (*Gardnerella vaginalis, Mobiluncus* sp., *Bacteroides* spp.) and aerobes in AV (group B streptococci, enterococci and *Escherichia coli*) are detected.

If an uncomplicated urogenital tract infection is diagnosed, empirical treatment is used. However, untreated or misdiagnosed vaginal infections can lead to pelvic inflammatory diseases and tubal infertility and ectopic pregnancy consequently, treatment of BV- and AV-associated bacteria may increase the vaginal reservoir of antibiotic-resistant strains. Polish data on the bacterial and fungal etiology of infections of the lower part of the reproductive system, let alone the antibiotic susceptibility, have not been collected. Meanwhile, implementing an antibiotic stewardship program (ASP) should be based on local data/local epidemiology: antibiograms and lab data, showing susceptibility of specific organisms to specific drugs in specific disease [4]. ASPs, by which we mean coordinated interventions designed to improve and measure the appropriate use of antibiotics by promoting the selection of the optimal pharmacotherapy, are also recommended in outpatient settings. However, the authors’ observations are quite different: antibiotic consumption in Poland is much higher than the European average [5]. Female patients are no exception, e.g., almost a quarter of Polish pregnant women and more than 7% of women in labor purchased antibiotics prescribed by the obstetrician [6,7]. Therefore, knowledge of the local epidemiology of infections should help physicians to conduct rational empirical antibiotic therapy.

The aim of the study was to analyze BV and AV associated bacteria and yeast strains isolated from vagina of women with and without symptoms of vaginal inflammation (pain, pruritus and dyspareunia) and vaginal discharge; and the results of antimicrobial susceptibility of the dominant bacterial and fungal etiological factors in outpatients of different age groups from the Silesian Voivodeship.

## 2. Materials and Methods

Retrospective laboratory-based multicenter study encompassed 4994 vaginal smears from women of different age in Silesian Voivodeship, the south of Poland. Patients who had vaginal swabs collected as per physicians’ orders during the period from 1 January 2017 to 30 June 2018 were included in the study. The inclusion criteria were non-hospitalized girls or women with and without symptoms of vaginal inflammation (pain, pruritus, dyspareunia) and vaginal discharge. Women excluded from the study were those over 80 years of age (*n* = 29), hospitalized, or suffering from typical sexually transmitted diseases, e.g., gonorrhoea, syphilis, chlamydia, trichomoniasis, human papillomavirus, genital herpes, HIV.

Vaginal discharge and swabs were collected using sterile cotton swabs, in lithotomy position, under aseptic conditions. These swabs were then transported to the diagnostic microbiology laboratory. Bacteria associated with AV and BV and fungi were identified using MALDI-TOF Biotyper (Biotyper Bruker Daltonics, Leiderdorp, The Netherlands) system according to the manufacturer’s instructions. The analysis of microbial etiology was carried out with stratification according to the age of patients. In as many as 18.8% of patients, rare species of bacteria were isolated, which were identified in not more than in 2% of all clinical samples—these were not subject to any further detailed analysis and are described in Table 1 as “other bacteria”.

All bacteria strains were tested using disk diffusion antimicrobial susceptibility methods on Mueller-Hinton agar plates according to current guidelines of the European Committee on Antimicrobial Susceptibility Testing (Clinical breakpoints tables v.6.0; http://www.eucast.org v.6.0), but cefoperazone/sulbactam according to the manufacturer’s instructions and the results were considered resistant (R) and susceptible (S), with intermediately resistant strains grouped together as drug-resistant. The initial antibiogram included antibiotics indicated in Table 1, for strains initially identified as drug-resistant, only ESBL-*E. coli* (extended-spectrum beta-lactamases producing *E. coli*) antibiogram was supplemented and they were: cefepim, ceftazidime and cefoperazone/sulbactam. All disks were obtained from Oxoid (Basingstoke, UK). The susceptibility of *Candida* spp. has not been tested [8].

The analysis of antibiotic susceptibility focused only on the most numerous microorganisms isolated (*Enterococcus faecalis*, *Escherichia coli* and *Streptoccoccus agalactiae*) and the age groups for which more than 5% of the total tests were performed, i.e., 15–54 years of age.

The database was compiled by the Chair of Microbiology at the Jagiellonian University Medical College (Cracow, Poland), and the collaborating laboratory from the Silesia region (Korlab Medical Laboratories, Ruda Śląska, Poland).

## 3. Results

The group of studied women was dominated by patients aged 25–44, which constituted 70.7% of all examined. The test was performed least often in girls up to 14 years of age (1.6%) and in this age group a negative test was most often obtained (37.8%, Table 1).

AV-associated bacteria such as Gram-positive cocci were the ones most often isolated: *Enterococcus faecalis* (29.2%) and *Streptoccoccus agalactiae* (13.1%), followed by bacilli from the *Enterobacteriaceae* group, including *Escherichia coli* (26.3%). The presence of *Streptococcus agalactiae* was confirmed in 13.1%, slightly more often in the 45–80 age group. BV associated *Gardnerella vaginalis* strains were detected in 6.4%, most often in women aged 15–24.

Yeast-like infections occurred in 24.3% of patients, more commonly in women aged 15–44, while in patients aged 55 and older, they accounted for only 12.5–10.1%, respectively (Table 1). *Candida albicans* accounted for 78.3%, whereas among *C.* non-*albicans* spp.—*C. glabrata* dominated (14.9%) followed by *C. parapsilosis* (3.8%).

In the studied age ranges, *Enterococcus faecalis* retained susceptibility to penicillins and aminoglycosides, although the high-level aminoglycoside resistance mechanism, HLAR, featured in 26.0% of strains.

Often isolated *Escherichia coli* rods showed high or medium susceptibility to aminoglycosides, less susceptibility to fluoroquinolones and trimethoprim/sulfamethoxazole. Of the antibiotics frequently used in genital tract infections, the lowest susceptibility was found to be with ampicillin. The ESBL mechanism was found in 3.6% of strains—these strains also showed very low susceptibility to third and fourth generation cephalosporins (Table 2).

*Streptococcus agalactiae* remained susceptible to penicillin and trimethoprim/sulfamethoxazole. Strains from patients aged 45–54 were found to be resistant to clindamycin and erythromycin (Table 2), and in general, the MLSB (Macrolide-lincosamide-streptogramin B) mechanism was identified in 38.6% of strains, mainly cMLSB (33.7%). (Table 2)

## 4. Discussion

It is puzzling that despite the fact that the vaginal flora and vaginal infections have been the topics of interest to physicians and scientists alike, for many years now, there is still much to be learned and newer scientific reports are continuing to improve our understanding of the vaginal flora and microbiology of infections. This is also the case in Poland, where such microbiology and antimicrobial resistance of infections are not described in detail, except for the phenomenon of *Streptococcus agalactiae* colonization among pregnant women [9,10].

*Streptococcus agalactiae* (GBS) is an important infectious agent in pregnant women and non-pregnant adults; in the studied population it was the fourth most-isolated microbe with a prevalence 13%. GBS can stimulate human cell lines in vitro (HeLa, THP-I and U 937) to release proinflammatory cytokines (IL-6, IL-8, TNF-α) [11]. However, above all, GBS emerged as the leading cause of neonatal sepsis, therefore, it is recommended to use either antenatal screening for GBS colonization and intrapartum antimicrobial prophylaxis (IAP) in colonized women or administer IAP to women with certain obstetric risk factors during labor. Penicillin is the first choice for intrapartum antibiotic prophylaxis, with ampicillin as an acceptable alternative for penicillin-allergic women at high risk for anaphylaxis: clindamycin [12]. Unfortunately, the presented data indicate high resistance to clindamycin and erythromycin, which is also confirmed by other Polish authors [13], but our data indicated also high resistance to ampicillin. Thus, a large proportion of GBS showing high drug resistance is a significant problem, of great importance to the reproductive health of Polish women, as evidenced by the polish Public Opinion Research Centre survey, according to which 41% of women aged 18–45 are planning offspring [14].

Numerous authors have indicated that the abnormal genital tract colonization, not only GBS, may lead to an in-utero inflammation/infectious process, and in pregnant women bacterial vaginosis may cause complications such as premature rupture of membranes and preterm delivery, respiratory distress syndrome or necrotizing enterocolitis (NEC) in preterm infants—especially *E. faecalis* (the most common microorganisms in our study) increased the risk for NEC [15]. Therefore, another important clinical problem in women of childbearing age is also AV, when owing to disappearance of *Lactobacillus* strains, mixed aerobic bacterial flora, mostly GBS, *Enterococcus* spp. and *E. coli* multiply. The development of purulent vaginal discharge in case of AV may be the cause of numerous complications, especially in pregnant women, which may predispose to infections from the vagina to higher sections of the reproductive system and premature rupture of membranes (PROM), miscarriages and premature delivery [16,17,18].

Microflora in desquamative inflammatory vaginitis usually consists of *E. coli, Staphylococcus aureus*, GBS and *Enterococcus faecalis* [19,20]. Unfortunately, our research found a high prevalence of *Escherichia coli*, which can colonize the vagina but also lead to the replacement of natural microflora, by eliminating lactobacilli. In cases of AV, as compared to GBS and *Enterococcus* spp., it is less often isolated, but during pregnancy it may be the cause of habitual miscarriages, chorioamnionitis and premature birth [21]. It can also increase the risk of early severe infections in a newborn [22,23]. In Poland, in very low birth weight neonates, *E. coli* constituted 12% of the pathogens isolated from neonatal sepsis and fatality case rate was 33% [24].

Treating bacterial vaginosis with different etiology during pregnancy can reduce poor outcomes, such as preterm birth, so WHO recommends not only treating symptomatic infections but also screening for vaginosis in pregnant women and women with a history of spontaneous abortion or preterm delivery [1,25], because it may be related to infertility. On the other hand, according to PREMEVA results, a double-blind randomized controlled trial done in 40 French facilities—screening and treatment of bacterial vaginosis in women without high-risk pregnancies shows no evidence of risk reduction of late miscarriage or spontaneous preterm birth [26].

Vaginal microbiota is essentially different during the individual periods in a woman’s life, and its species composition depends mainly on hormone management, sexual activity and the efficient functioning of local immune mechanisms, associated with mucosa-associated lymphoid tissue (MALT), as well as systemic mechanisms. In the analyzed group of sexually active outpatient patients of childbearing age, up to 45 years, *Candida albicans* was the most frequently isolated fungus, which caused lower genital tract infections in about a third of women aged 15–24. Moreover, *Gardnerella vaginalis* strains were isolated from women, whose participation in the pathogenesis, both qualitative and quantitative disorders of the vaginal microbiota, with the disappearance of *Lactobacillus* strains (resulting in the formation of BV)—and specific discharge decreases significantly after the age of 54.

Vaginal infections caused by *Candida* spp. and BV are a therapeutic problem in the group of outpatients, due to their frequent recurrent character. These infections may impair the functioning of immune mechanisms, favoring the promotion and progression of HPV-dependent dysplastic changes in the cervical epithelium. In the studies of Ekiel et al. in women with an average age of 38 years with the diagnosis of cytological changes of ASCUS type (atypical squamous cells of undetermined significance), showed *Candida albicans* strains and no lactobacilli were isolated. In comparison to the control group of 30-year-old women without any dysplastic lesions of the cervical epithelium, GBS strains and gram-negative bacilli were also more often isolated from the vagina [27].

Unfortunately, vulvovaginal candidiasis may also be related to infertility problems [28] and we observed a very high level of yeast infections, about a quarter of all had such aetiology, certain medical conditions, e.g., pregnant women and women using oral contraceptives (changes in vaginal acidity, pH) or diabetes may increase the risk of yeast infections. Less commonly, recurrent yeast infections may be a sign of a more serious illness that reduces immunity (such as long-term chronic illness or HIV infection). These should be considered only if there are other symptoms; yeast infection alone is common and usually easily prevented or treated.

Treatment of lower reproductive tract infections includes the administration of antibiotics, such as nitroimidazoles, lincosamides, macrolides (e.g., erythromycin) and, in some circumstances, penicillins [8], or antifungal drugs, such as fluconazole or amphotericin B [29]. Pharmacotherapy of infections depends on the drug susceptibility of microorganisms, and on the bioavailability of drugs. Distribution to the body’s tissues after a drug enters the systemic circulation, is generally uneven because of differences in blood perfusion, tissue binding (e.g., because of lipid content), regional pH, and permeability of cell membranes. Unfortunately, the bioavailability of antibiotics or antifungal drugs in vaginal tissues cannot always be predicted and can be much lower, hence the main regiment of the metronidazole or clindamycin, clotrimazole, miconazole, and tioconazole preparations is intravaginal [29,30,31], and the recommended route of admission of other antibiotics is oral. According to Sobel et al., the susceptibility testing for women with vulvovaginal candidiasis appears to be unjustified [29], whereas antibiotics should be taken after drug-susceptibility testing. Data from clinical trials indicate that a woman’s response to the therapy of bacterial vaginosis, and the likelihood of relapse or recurrence, are not affected by treatment of her sex partner(s), but routine treatment of sex partners is recommended in vulvovaginal candidiasis [8].

The presented research results also show another additional aspect of everyday practice in gynaecological proceedings: diagnostics of lower genital tract infections. Vaginal infections should be diagnosed using both clinical criteria—Amsel test—as a screening test for BV diagnostics and microbiological Nugent criteria as a confirmation test with inoculation in doubtful situations or lack of effect after initial treatment [32]. Unfortunately, in the analyzed cases only the Amsel criteria and culture methods were used, omitting the preparations that were not ordered by the physicians-in-charge, and the experience of the authors indicate that this may be a problem not only in Silesian province, but in diagnostics of lower urinary tract infections in Poland. This is the more significant problem, that a large variety of etiological factors of genital tract infections (in our studies, from 18.8% of patients, we isolated rare species of microorganisms) indicates the necessity of using better and better methods of identification, not only morphotypes in microscopic examination (e.g., *Peptostreptococcus* spp., *Atopobium vaginae* described in the literature), but also studies of non-microbial microorganisms (e.g., *Atopobium vaginae*) [33].

This retrospective laboratory-based study had some limitations. Demographic information for the study population was limited, thus, data on the use of contraceptives, sexual activity and partners, sign and symptoms, such as vulvar pruritus, were unavailable, as well as signs of atypical infections like cervicitis. Despite the large population constituting the study group, the limitation of the study is also the fact that it only covers one Silesia province. The most important limitation of the study was not using samples for Nugent scoring.

## Figures and Tables

**Table 1 ijerph-18-00335-t001:** Distributions of the microorganisms in infections of the female reproductive tract in women from the south of Poland.

Microorganisms	Age of Patients (Years) *n* (%)									Range
	≤14	15–24	25–34	35–44	45–54	55–64	65–80	BD	Total	
Gram-positive										
*Staphylococcus aureus*	0 (0.0)	23 (4.3)	62 (2.8)	25 (1.9)	7 (1.9)	9 (4.0)	6 (2.8)	1 (3.8)	135 (2.7)	8
*Enterococcus faecalis*	26 (31.7)	117 (21.8)	651 (28.9)	360 (27.8)	121 (32.9)	80 (35.7)	96 (44.0)	7 (26.9)	1458 (29.2)	1
*Enterococcus faecium*	0 (0.0)	0 (0.0)	2 (0.1)	1 (0.1)	1 (0.3)	0 (0.0)	1 (0.5)	0 (0.0)	5 (0.1)	10
*Enterococcus* spp.	2 (2.4)	0 (0.0)	1 (0.0)	0 (0.0)	0 (0.0)	1 (0.4)	3 (1.4)	0 (0.0)	7 (0.1)	10
*Streptococcus agalactiae*, GBS	2 (2.4)	76 (14.2)	274 (12.2)	160 (12.3)	65 (17.7)	41 (18.3)	37 (17.0)	2 (7.7)	657 (13.1)	4
*Gardnerella vaginalis*	0 (0.0)	55 (10.3)	158 (7.0)	72 (5.6)	22 (6.0)	11 (4.9)	2 (0.9)	2 (7.7)	322 (6.4)	5
Gram-negative										
*Escherichia coli*	39 (47.6)	113 (21.1)	553 (24.5)	298 (23.0)	115 (31.3)	80 (35.7)	106 (48.6)	6 (23.1)	1309 (26.2)	2
*Klebsiella pneumoniae*	4 (4.9)	21 (3.9)	94 (4.2)	49 (3.8)	15 (4.1)	12 (5.4)	14 (6.4)	0 (0.0)	209 (4.2)	7
*Klebsiella oxytoca*	0 (0.0)	0 (0.0)	11 (0.5)	2 (0.2)	3 (0.8)	5 (2.2)	3 (1.4)	1 (3.8)	25 (0.5)	8
*Klebsiella mobilis*	1 (1.2)	0 (0.0)	8 (0.4)	4 (0.3)	0 (0.0)	1 (0.4)	2 (0.9)	0 (0.0)	16 (0.3)	9
*Klebsiella* spp.	0 (0.0)	0 (0.0)	1 (0.0)	0 (0.0)	0 (0.0)	0 (0.0)	0 (0.0)	0 (0.0)	1 (0.01)	11
other bacteria *	31 (37.8)	95 (17.7)	422 (18.7)	220 (17.0)	61 (16.6)	40 (17.9)	61 (28.0)	3 (11.5)	933 (18.7)	
*Candida* spp.	7 (8.5)	173 (32.3)	564 (25.0)	339 (26.2)	80 (21.7)	28 (12.5)	22 (10.1)	6 (23.1)	1219 (24.4)	3
Negative result	31 (37.8)	160 (29.9)	665 (29.5)	410 (31.6)	133 (36.1)	82 (36.6)	65 (29.8)	9 (34.6)	1555 (31.1)	
Total no. of patients	82 (100)	536 (100)	2253 (100)	1296 (100)	368 (100)	224 (100)	218 (100)	26 (100)	4994 (100%)	

* The species or families of strains whose isolation was over 100.

**Table 2 ijerph-18-00335-t002:** Antimicrobial susceptibility of selected microorganisms isolated from infections according to patients’ age.

Antimicrobial/Strain	Patient Age (Years)				
	15–24	25–34	35–44	45–54	Total
*Enterococcus faecalis* susceptibility (%)					
Ampicillin (5 µg)	100.0	100.0	100.0	100.0	100
Gentamicin High Level (synergy) (30 µg)	79.2	81.6	78.4	78.4	79.7
*Streptococcus agalactiae*, GBS susceptibility (%)					
Benzylpenicillin (1 unit)	100.0	100.0	100.0	100.0	100.0
Clindamycin (2 µg)	71.1	72.4	67.3	51.5	69.2
Erythromycin (15 µg)	68.4	69.5	64.8	53.0	66.9
Trimethoprim/ Sulfamethoxazole (1.25/23.75 µg)	98.7	98.9	98.1	97.0	98.6
*Escherichia coli* susceptibility (%)					
Aminoglycosides					
Gentamicin (10 µg)	95.6	96.2	95.4	93.9	95.0
Tobramycin (10 µg)	100.0	61.9	90.9	66.7	68.2
Amikacin (30 µg)	100.0	85.7	100.0	88.9	87.9
Non-extended spectrum cephalosporins; 1st and 2nd generation cephalosporins					
Cefuroxime (30 µg)	97.3	96.0	97.0	95.7	96.0
Extended-spectrum cephalosporins; 3rd and 4th generation cephalosporins					
Cefotaxime (5 µg)	97.3	97.1	97.3	95.7	96.8
Ceftazidime * (*n* = 66) (10 µg)	0.0	50.0	54.5	62.5	54.0
Cefepime * (*n* = 66) (30 µg)	50.0	45.5	45.5	44.4	47.0
Cefoperazone/Sulbactam * (*n* = 66) (75/30 µg)	66.7	100.0	100.0	100.0	98.4
Fluoroquinolones					
Ciprofloxacin (5 µg)	84.6	85.2	85.2	83.6	82.5
Folate pathway inhibitors					
Trimethoprim/Sulfamethoxazole (1.25/23.75 µg)	81.4	79.4	80.1	70.1	76.6
Penicillins					
Ampicillin (10 µg)	53.5	47.9	51.9	48.8	49.0
Piperacillin/Tazobactam (30/6 µg)	100.0	90.5	100.0	100.0	90.9
Penicillins + b-lactamase inhibitors					
Ampicillin/Sulbactam (10/10 µg)	86.0	80.3	85.0	78.5	81.1

* Studied only in drug resistant strains, ESBL-*E. coli.*

## Data Availability

The datasets during and/or analysed during the current study available from Jadwiga Wojkowska-Mach (e-mail: mbmach@cyf-kr.eu.pl) on reasonable request

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
