# Peer review of "Prevalence and Antimicrobial Susceptibility Profiles of Microorganisms Associated with Lower Reproductive Tract Infections in Women from Southern Poland—Retrospective Laboratory-Based Study"

_ijerph, 2021, doi:10.3390/ijerph18010335_

Round 1

Reviewer 1 Report

The goal of the paper is to explore the microbiology of the vagina using a retrospective analysis.

Specific points:

  1. The small paragraph from lines 37 to 40 would be most effective as part of the introduction or not included in the paper.
  2. The introduction lacks specific details and is very general. It does not make a strong case for the purpose of the study.
  3. The methods does not describe whether there as exclusion of the data or how the participants were selected.

The paper requires some editing for grammar.

Author Response

DETAILED RESPONSE TO REVIEWER

STEP-BY-STEP REPLIES TO REVIEWERS' COMMENTS:

As suggested, we have introduced the necessary changes according to reviewers’ suggestions and comments. In our opinion, the readability of the manuscript has greatly improved, due to the suggestions from reviewer. Enclosed with the cover letter, please find the point-by-point replies to the reviewers’ comments. All changes in revised manuscript were yellow highlighted.

Reviewer #1:

The goal of the paper is to explore the microbiology of the vagina using a retrospective analysis. Specific points:

The small paragraph from lines 37 to 40 would be most effective as part of the introduction or not included in the paper.

Authors’ reply: Corrected according to suggestions (lines 37-40 belong to the “Introduction” section)..

The introduction lacks specific details and is very general. It does not make a strong case for the purpose of the study.

Authors’ reply: Corrected according to suggestions (line 61-72), as below:

“(...) Polish data on the bacterial and fungal aetiology of infections of the lower part of the reproductive system, let alone the antibiotic susceptibility, have not been heard. Meanwhile, implementing an antibiotic stewardship programme (ASP) should be based on local data / local epidemiology: antibiograms and lab data, showing susceptibility of specific organisms to specific drugs in specific disease [4]. ASPs, by which we mean coordinated interventions designed to improve and measure the appropriate use of antibiotics by promoting the selection of the optimal pharmacotherapy, are also recommended in outpatient settings. However, the authors’ observations are quite different: antibiotic consumption in Poland is much higher than the European average [5]. Female patients are no exception, e.g. almost a quarter of Polish pregnant women and more than 7% of women in labour purchased antibiotics prescribed by the obstetrician [6,7]. Therefore, knowledge of the local epidemiology of infections should help physicians to conduct rational empirical antibiotic therapy. (...)”

The methods does not describe whether there as exclusion of the data or how the participants were selected.

Authors’ reply: Corrected according to suggestions, the “Materials and methods” section was supplemented (line 84-86).

The paper requires some editing for grammar.

Authors’ reply: Corrected according to suggestions.

Reviewer 2 Report

More detail report is attached but general observations are:

  1. Journal References style was not followed.
  2. The authors mixed UK and America English together. There is need to adhere to uniformity in English Language style.  

Author Response

DETAILED RESPONSE TO REVIEWER

STEP-BY-STEP REPLIES TO REVIEWERS' COMMENTS:

As suggested, we have introduced the necessary changes according to reviewers’ suggestions and comments. In our opinion, the readability of the manuscript has greatly improved, due to the suggestions from reviewer. Enclosed with the cover letter, please find the point-by-point replies to the reviewers’ comments. All changes in revised manuscript were yellow highlighted.

Reviewer #2:

More detail report is attached but general observations are, Journal References style was not followed.

Authors’ reply: Corrected according to suggestions.

The authors mixed UK and America English together. There is need to adhere to uniformity in English Language style.  

Authors’ reply: Corrected according to suggestions.

Reviewer 3 Report

The article by Mach et al studies antimicrobial profiles of microorganisms associated with lower reproductive tract infections in women. Most important concern about the study is that the rationale is not clear from the introduction. What are the significances of studying the microbial profiles of patients suffering from vaginal infections? In other words how the study will help in therapeutics?

In table 2 no antifungal agents like fluconazole or amphotericin B were used to analyze resistance of the yeast isolates from the patients since multi drug resistant vaginal yeast infection with candida are common as showed by the following references:

https://aac.asm.org/content/47/1/34

https://aac.asm.org/content/60/10/5858.short

https://link.springer.com/article/10.1007/s11908-001-0093-5

https://www.mdpi.com/2079-6382/9/6/312

Susceptibility profiles of the yeast isolates need to be analyzed.

As mentioned in the above references, bioavailability of antifungal drugs are much lower in vaginal tissues, is it true for antibiotics? If so that may be problematic in determining the susceptibility profile from in vitro studies as the authors have done. Please explain.

Several places english grammar is off. For example, line 16 "the aim of the study was to analyze of ethology..." should be "... study was to analyze ethology..

Lines 37-40 belongs to which section? Please clarify

Author Response

DETAILED RESPONSE TO REVIEWER
STEP-BY-STEP REPLIES TO REVIEWERS' COMMENTS:
As suggested, we have introduced the necessary changes according to reviewers’ suggestions and comments. In our opinion, the readability of the manuscript has greatly improved, due to the suggestions from reviewer. Enclosed with the cover letter, please find the point-by-point replies to the reviewers’ comments. All changes in revised manuscript were yellow highlighted. 

Reviewer #3:
The article by Mach et al studies antimicrobial profiles of microorganisms associated with lower reproductive tract infections in women. Most important concern about the study is that the rationale is not clear from the introduction. What are the significances of studying the microbial profiles of patients suffering from vaginal infections? In other words how the study will help in therapeutics?
Authors’ reply: Corrected according to suggestions (line 61-72), as below: 

“(...) Polish data on the bacterial and fungal aetiology of infections of the lower part of the reproductive system, let alone the antibiotic susceptibility, have not been heard. Meanwhile, implementing an antibiotic stewardship programme (ASP) should be based on local data / local epidemiology: antibiograms and lab data, showing susceptibility of specific organisms to specific drugs in specific disease [4]. ASPs, by which we mean coordinated interventions designed to improve and measure the appropriate use of antibiotics by promoting the selection of the optimal pharmacotherapy, are also recommended in outpatient settings. However, the authors’ observations are quite different: antibiotic consumption in Poland is much higher than the European average [5]. Female patients are no exception, e.g. almost a quarter of Polish pregnant women and more than 7% of women in labour purchased antibiotics prescribed by the obstetrician [6,7]. Therefore, knowledge of the local epidemiology of infections should help physicians to conduct rational empirical antibiotic therapy. (...)”
In table 2 no antifungal agents like fluconazole or amphotericin B were used to analyze resistance of the yeast isolates from the patients since multi drug resistant vaginal yeast infection with candida are common as showed by the following references: https://aac.asm.org/content/47/1/34 https://aac.asm.org/content/60/10/5858.short https://link.springer.com/article/10.1007/s11908-001-0093-5 https://www.mdpi.com/2079-6382/9/6/312 Susceptibility profiles of the yeast isolates need to be analyzed.

As mentioned in the above references, bioavailability of antifungal drugs are much lower in vaginal tissues, is it true for antibiotics? If so that may be problematic in determining the susceptibility profile from in vitro studies as the authors have done. Please explain.
Authors’ reply: Corrected according to suggestions ( lines 103-104 and lines 215-229), as below: 

“(...) The susceptibility of Candida spp. has not been tested [8]. (...) Treatment of lower reproductive tract infections includes the administration of antibiotics, such as nitroimidazoles, lincosamides, macrolides (e.g., erythromycin) and, in some circumstances, penicillins [8], or antifungal drugs, such as fluconazole or amphotericin B [29]. Pharmacotherapy of infections depends on the drug susceptibility of microorganisms, and on the bioavailability of drugs. Distribution to the body’s tissues after a drug enters the systemic circulation, is generally uneven because of differences in blood perfusion, tissue binding (e.g., because of lipid content), regional pH, and permeability of cell membranes. Unfortunately, the bioavailability of antibiotics or antifungal drugs in vaginal tissues cannot always be predicted and much lower, hence the main regiment of the metronidazole or clindamycin, clotrimazole, miconazole, and tioconazole preparations is intravaginal [29,3031], recommended route of admission of other antibiotics is oral. According to Sobel et al., the susceptibility testing for women with vulvovaginal candidiasis appears to be unjustified [29], whereas antibiotics should be taken after drug-susceptibility testing. Data from clinical trials indicate that a woman's response to the therapy of bacterial vaginosis, and the likelihood of relapse or recurrence, are not affected by treatment of her sex partner(s), but routine treatment of sex partners is recommended in vulvovaginal candidiasis [8]. (...)”

Several places English grammar is off. For example, line 16 "the aim of the study was to analyze of ethology..." should be "... study was to analyze ethology..
Authors’ reply: Corrected according to suggestions. 

Lines 37-40 belongs to which section? Please clarify
Authors’ reply: Corrected according to suggestions (lines 37-40 belong to the “Introduction” section). 

Round 2

Reviewer 1 Report

The authors have addressed major concerns.

Reviewer 3 Report

The authors have satisfactorily answered to all my queries.